Characteristics and risk factors of infections in patients with HBV-related acute-on-chronic liver failure: a retrospective study

Zhang Qian 1 bearaly@126.com
Shi Baoxian 2
Wu Liang 3 wutongji@hotmail.com
1 Division of Nephrology, Department of Internal Medicine, Tongji Hospital, Tongji Medical College, Huazhong University of Science and Technology , Wuhan , China
2 Department of Chemistry and Environmental Engineering, Wuhan Polytechnic University , Wuhan , China
3 Department and Institute of Infectious Disease, Tongji Hospital, Tongji Medical College, Huazhong University of Science and Technology , Wuhan , China
Suner Aslı
Electronic publication date: 2022 Jul 5
Publication date: 2022
Volume: 10
Electronic Location ID: e13519
Received 2021 Nov 30; Accepted 2022 May 9
Copyright: © 2022 Zhang et al.
Copyright year: 2022
Copyright holder: Zhang et al.
License: This is an open access article distributed under the terms of the Creative Commons Attribution License, which permits unrestricted use, distribution, reproduction and adaptation in any medium and for any purpose provided that it is properly attributed. For attribution, the original author(s), title, publication source (PeerJ) and either DOI or URL of the article must be cited.
License URL: https://creativecommons.org/licenses/by/4.0/

Keywords: Hepatitis B virus, Acute-on-chronic liver failure, Infection, Risk factors

Funding: National Natural Science Foundation of China 81500546 This work was supported by grants (No. 81500546) from the National Natural Science Foundation of China. The funders had no role in study design, data collection and analysis, decision to publish, or preparation of the manuscript.

==============================
Background

Acute-on-chronic liver failure (ACLF) is a syndrome characterized by acute decompensation, organ failures, and high short-term mortality whose main cause in China is the Hepatitis B virus (HBV). Moreover, one of the most important causes of morbidity and mortality in HBV-ACLF patients is bacterial infection. Therefore, we investigate the clinical features, risk factors, prophylaxis and management of infections in patients with HBV-ACLF.

Methods

We conducted a retrospective analysis of 539 patients with HBV-ACLF in Wuhan Tongji Hospital from October 2015 to May 2018. Differences among groups were compared with Student’s t test, Mann–Whitney U test, χ2 test, or Fisher exact test as appropriate. Univariate and Multivariate logistic regression analysis was used for modeling the relationship between infection and clinical characteristics of HBV-ACLF.

Results

In total 58.81% (317/539) of patients with HBV-ACLF became complicated with infections, and the most common types were spontaneous bacterial peritonitis, urinary tract infection and pulmonary infection. Additionally, 32.18% (102/317) of patients suffered multi-organ infections, and 95.73% (516/539) of patients received anti-infective therapy. We detected a total of 202 isolates in all infected patients, and Escherichia coli (36.14%, 73/202) was the most common causative organism. Moreover, antibiotic susceptibility test patterns showed that 52.97% (107/202) of pathogens were MDR bacteria and 4.95% (10/202) were XDR bacteria. Univariate analysis indicated that patients with infection had a higher proportion of females, taking alcohol, diuretics, hepatic encephalopathy (HE), hepatorenal syndrome (HS), cirrhosis, a long-time in bed and mechanical ventilation, lower prothrombin activity (PTA), alanine aminotransferase (ALT), albumin, total cholesterol (TC), estimated glomerular filtration rate (eGFR), hemoglobin (Hb) and platelet (PLT) and higher age, model for end-stage liver disease (MELD) scores and ACLF grade than patients without infection. Multivariate logistic regression analysis showed that taking alcohol, HE, HS, cirrhosis, albumin and eGFR were risk factors for the development of infection.

Conclusions

Bacterial infections were very common in patients with HBV-ACLF. Taking alcohol, the occurrence of complications (HE, HS and cirrhosis), hypoalbuminemia and poor renal function often predict the higher prevalence of infections in patients with HBV-ACLF. It is important to focus on exploring the early recognition of infection and early intervention of those risk factors in patients with HBV-ACLF.

Introduction

Liver failure is a serious acute or chronic liver insufficiency induced by a variety of causes, which can result in obstruction or decompensation of liver function including synthesis, detoxification, excretion and biotransformation functions, and then can be further followed by clinical manifestations, such as coagulation disorders, jaundice, hepatic encephalopathy, ascitic fluid (Xie et al., 2016). Hepatitis B virus (HBV) infection is the main cause of liver failure in China. Liver failure can be classified into acute liver failure (ALF), subacute liver failure (SALF), acute-on-chronic liver failure (ACLF) and chronic liver failure (Sarin et al., 2019). In particular, ACLF is a clinical syndrome of acute liver decompensation that results from a precipitating event (PE) in patients with previously compensated liver disease (Jalan et al., 2012). In the West, acute alcohol injury and bacterial infections are the most common types. In the East, this is true also for India and Korea (Chen et al., 2018). However, in China, HBV-related ACLF is the leading cause of liver failure because of the high incidence of chronic HBV infection (You et al., 2013; Gu et al., 2018). Patients with liver failure often also suffer from immunodeficiency, which increases the risk and severity of infection, including spontaneous bacterial peritonitis (SBP), pulmonary infection, urinary tract infection (UTI) and even sepsis. It can seriously affect the prognoses of patients with HBV-ACLF (Leber, Spindelboeck & Stadlbauer, 2012). Meanwhile, bacterial infections can also trigger the occurrence of ACLF in patients with cirrhosis (Li et al., 2020). However, clinicians still face the important questions of how to prevent the occurrence and development of HBV-ACLF and how to treat and predict the prognosis of HBV-ACLF patients effectively every day. Therefore, the aim of this study is to evaluate the prevalence and types of infection, characteristics of multiple infections and related risk factors in patients with HBV-ACLF. We hope to propose strategies to prevent the development of HBV-ACLF and improve patients’ qualities of life.

Methods

Ethics approval and consent to participate

This research project was performed in accordance with the principles of the Helsinki Declaration. The Ethics Research Committee at Tongji Hospital approved the study (TJ-IRB20190601) and agreed that patient consent was not needed. The information of all patients was anonymized and de-identified prior to analysis.

Participants and design

Patients with HBV-ACLF who were hospitalized at the Department and Institute of Infectious Disease of Tongji Hospital in Wuhan of China from October 2015 to May 2018 were selected from the electronic database for retrospective analysis using the following inclusion criteria. First, all patients had to be more than 18 years old; second, all patients had to meet the consensus recommendations of the Asian Pacific association for the study of the liver (APASL). Data were collected as previously described in Sarin et al. (2019). Specifically, jaundice (serum bilirubin ≥5 mg/dL (85 mmol/L) and coagulopathy (INR ≥1.5 or prothrombin activity <40%) complicated by clinical ascites and/or encephalopathy with previously diagnosed or undiagnosed chronic liver disease/cirrhosis within 4 weeks, and is associated with a high 28-day mortality (Sarin et al., 2019).

We excluded patients from the study based on the following. (1) Coinfection with other viruses (hepatitis A, C, D or E), human immunodeficiency virus (HIV), or syphilis; (2) Hepatic disease in schistosomiasis, alcoholic hepatic disease, drug-induced hepatitis, fatty liver disease, autoimmune liver disease, Wilson disease or hepatocellular carcinoma; (3) SLE, pregnant patients and postoperative patients; (4) the instrumentation of patients with urinary catheters; (5) Patients had no record of urine routine.

Clinical and laboratory data collection

Information regarding the age, gender, body mass index (BMI), course of hepatitis, types of infections, medication used (including diuretics and anti-infective drugs), severe complications (including hepatic encephalopathy, liver cirrhosis and hepatorenal syndrome) and laboratory tests were recorded. Laboratory parameters included the copies of HBV DNA, alanine aminotransferase (ALT), aspartate transaminase (AST), total bilirubin (TBIL), direct bilirubin (DBIL), albumin, globulin, total cholesterol (TC), triglycerides (TG), prothrombin activity (PTA), glomerular filtration rate (GFR), white blood cell (WBC) count, hemoglobin (Hb) and platelet (PLT).

We deemed various bacterial infections to be present using the following criteria (Mucke et al., 2018; Yokoe et al., 2018; Li et al., 2020): (1) SBP: ascetic fluid >250/mL polymorphonuclear cells; (2) pulmonary infection: a combination of respiratory symptoms (cough, sputum, dyspnea, pleuritic pain), typical findings on auscultation (rales or crepitation), signs of infection (fever, elevated WBC) and new pulmonary infiltrate; (3) UTI: a combination of urine white blood count >11 cells/μL in males and 35 cells/μL in females with positive culture or uncountable leucocytes per field if they were negative cultures; (4) cholecystitis: local signs of inflammation (Murphy’s sign, right upper quadrant tenderness/pain/mass), systemic signs of inflammation (fever, elevated CRP, elevated WBC), and positive imaging findings (sonographic Murphy’s sign, gallbladder wall thickening, pericholecystic fluid/oedema); (5) other infections: including appendicitis, perianal abscess, cytomegalovirus, herpes zoster, viral conjunctivitis, EB virus or skin/soft tissue infections, etc. We defined multidrug-resistant (MDR) bacteria as those bacteria with non-susceptibility to at least one agent in three or more antimicrobial categories and extensively-drug resistance (XDR) bacteria as those with non-susceptibility to at least one agent in all but two or fewer antimicrobial categories (Magiorakos et al., 2012).

In addition to the above, we calculated Model for End-stage Liver Disease (MELD) scores for all patients at admission. Organ failure and ACLF were classified according to the following specific criteria. For EASL-CLIF organ system failure, (1) liver: serum bilirubin ≥12 mg/dL; (2) coagulation: INR ≥2.5 or platelet count ≤20 × 109/L; (3) kidney: serum creatinine ≥2 mg/dL or the need for dialysis; (4) brain: grade III-IV HE based on West Haven criteria; (5) circulation: treatment with vasoconstrictors to maintain arterial pressure or inotropes to improve cardiac output; (6) lung: PaO2/FiO2 ≤ 200 or SpO2/FiO2 ≤ 214 or need for mechanical ventilation. EASL-CLIF ACLF definition and grading: ACLF grade 1 (ACLF-1): kidney failure or other single organ failure if associated with kidney dysfunction (serum creatinine ranging from 1.5 to 1.9 mg/dL) and/or grade I–II HE; ACLF-2 was defined by the presence of 2 organ failure and ACLF-3 for 3 or more organ failures (Cao et al., 2020).

Power analysis

The G * Power software (latest ver. 3.1.9.7; Heinrich-Heine-Universität Düsseldorf, Düsseldorf, Germany; http://www.gpower.hhu.de) was used to perform the power analysis. The sample size calculated for the values (α = 0.05, power = 1 − β = 0.9, effect size = 0.26 according to Cohen suggestion) with a priori analysis was N = 348. As the total sample size N = 539 was given after the completion of the study, the corresponding power was calculated as “0.9827108” with a post-hoc analysis. The required population effect size was calculated as “0.21” with sensitivity analysis (Faul et al., 2009; Serdar et al., 2021).

Data analysis

All statistical analysis was performed using the IBM SPSS 20.0 software. Continuous variables were tested for normal distributions with the Shapiro test. The categorical data were reported as numbers and percentages. Mean ± SD and median (interquartile range [IQR]) were used for describing normal and skewed distributed variables, respectively. Differences among groups were compared with Student’s t test, Mann–Whitney U test, χ2 test, or Fisher exact test as appropriate. Univariable analysis was performed to identify independent predictors of infections and only candidate variables (P < 0.2) were entered as potential covariates into the multivariate model. Multivariable logistic regression analysis (Forward: LR) was used for modeling the relationship between infection and clinical characteristics. The P-values reported were two-sided and taken to be significant at <0.05.

Results

Patient characteristics

After applying the inclusion and exclusion criteria mentioned above, we included a total of 539 patients with HBV-ACLF in this study. Among this cohort, 317 patients (58.81%) were complicated with infections and 222 patients (41.19%) had no infections. We then compared the baseline demographics, reasons for admission, clinical and laboratory data, MELD scores and ACLF grades between HBV-ACLF patients with and without infection (Table 1).

Table 1 Baseline demographic, clinical, and laboratory characteristics of the study population.

	All (n = 539)	Infection group (n = 317)	Non-infection group (n = 222)	P Value	
Age (years)a	44.88 ± 11.50	46.69 ± 11.22	42.31 ± 11.44	P < 0.001	
Male, n (%)	468 (86.83%)	267 (84.23%)	201 (90.54%)	0.033	
Smoker, n (%)	239 (44.34%)	145 (45.74%)	94 (42.34%)	0.434	
Drinker, n (%)	188 (34.88%)	122 (38.49%)	66 (29.73%)	0.036	
BMI (kg/m2)	23.18 (20.76–25.26)	23.01 (20.76–25.26)	23.18 (20.76–25.61)	0.786	
Course of hepatitis (years)	9 (2–17)	9 (2–20)	9 (1.44–14.44)	0.196	
Admission reason, n (%)					
Ascites	103 (19.11%)	82 (25.87%)	22 (9.91%)	P < 0.001	
GEVB	6 (1.11%)	6 (1.89%)	0 (0%)	0.100	
HE	24 (4.45%)	13 (4.1%)	11 (4.95%)	0.636	
Infection	92 (17.07%)	92 (29.02%)	0 (0%)	P < 0.001	
Virus replication	46 (8.53%)	24 (7.57%)	22 (9.91%)	0.339	
Jaundice	268 (49.72%)	100 (31.55%)	167 (75.23%)	P < 0.001	
Diuretics, n (%)					
Negative	148 (27.46%)	67 (21.14%)	81 (36.49%)	–	
Positive	391 (72.54%)	250 (78.86%)	141 (63.51%)	P < 0.001	
HE, n (%)					
Negative	416 (77.18%)	226 (71.29%)	190 (85.59%)	–	
Positive	123 (22.82%)	91 (28.71%)	32 (14.41%)	P < 0.001	
HS, n (%)					
Negative	460 (85.34%)	247 (77.92%)	213 (95.95%)	–	
Positive	79 (14.66%)	70 (22.08%)	9 (4.05%)	P < 0.001	
Cirrhosis, n (%)					
Negative	347 (64.38%)	181 (57.10%)	166 (74.77%)	–	
Positive	192 (35.62%)	136 (42.90%)	56 (25.23%)	P <0.001	
Long-time bed, n (%)					
Negative	282 (52.32%)	148 (46.69%)	134 (60.36%)	–	
Positive	257 (47.68%)	169 (53.31%)	88 (39.64%)	0.002	
Mechanical ventilation, n (%)	17 (3.15%)	16 (5.05%)	1 (0.45%)	0.003	
Artificial liver, n (%)	314 (58.26%)	174 (54.89%)	140 (63.06%)	0.058	
Laboratory data at admission					
PTA (%)	35 (27–40)	33 (27–39)	37 (29–40)	0.002	
INR	2.28 (2–2.83)	2.36 (2.01–2.9)	2.16 (1.99–2.73)	0.121	
ALT (U/L)	280 (87–713)	214 (71.5–597)	358.5 (130.75–772)	0.001	
AST (U/L)	205 (108–490)	190 (101.5–456.5)	215 (117.75–508.75)	0.187	
TBIL (μmol/L)a	300.59 ± 137.81	308.03 ± 150.79	289.98 ± 116.32	0.118	
DBIL (μmol/L)a	214.21 ± 100.13	219.76 ± 109.69	206.28 ± 84.24	0.108	
ALB (g/L)a	30.90 ± 4.98	29.68 ± 4.61	32.70 ± 4.64	P < 0.001	
Globulin (g/L)a	28.42 ± 7.58	28.36 ± 7.70	249 ± 7.39	0.838	
TC (mmol/L)	2.19 (1.69–2.74)	2.14 (1.59–2.65)	2.28 (1.83–2.86)	0.003	
TG (mmol/L)	1.02 (0.85–1.28)	1.01 (0.84–1.25)	1.04 (0.85–1.33)	0.174	
eGFR (ml/min/1.73 m2)	117.1 (100.26–129.51)	111.98 (86.94–125.66)	123.43 (109.14–133.27)	P < 0.001	
WBC (×109/L)	6.11 (4.49–8.41)	6.03 (4.47–8.79)	6.19 (4.62–7.76)	0.715	
Hb (g/L)	127 (109–140)	124 (105.5–138)	130.5 (113.75–142)	P < 0.001	
PLT (×109/L)	104 (71–139)	100 (62.5–134)	109 (76.75–149)	0.004	
HBV DNA (log10 IU/ml)	5.19 (3.48–6.70)	5.16 (3.28–6.55)	5.3 (3.74–6.92)	0.08	
HBeAg positive, n (%)	178 (33.02%)	102 (32.18%)	76 (34.23%)	0.617	
MELD scores	27 (25–30)	27 (23–31)	26 (23–29)	0.003	
ACLF grade, n (%)					
grade 1	317 (58.81%)	165 (52.05%)	152 (68.47%)	P < 0.001	
grade 2	128 (23.75%)	83 (26.18%)	45 (20.27%)	0.112	
grade 3	94 (17.44%)	69 (21.77%)	25 (11.26%)	0.002	
Notes:

ACLF, acute-on-chronic liver failure; ALB, albumin; ALT, alanine aminotransferase; AST, aspartate transaminase; BMI, body mass index; DBIL, direct bilirubin; eGFR, estimated glomerular filtration rate; GEVB, gastroesophageal variceal bleeding; Hb, hemoglobin; HBeAg, Hepatitis B virus e antigen; HBV, hepatitis B virus; HE, hepatic encephalopathy; HS, hepatorenal syndrome; MELD, model for end-stage liver disease; PLT, platelet; PTA, prothrombin activity; TBIL, total bilirubin; TC, total cholesterol; TG, triglyceride; WBC, white blood cell. Values are expressed as median (interquartile range) unless otherwise specified.

a Values are expressed as means ± SD.

Prevalence and types of infection in patients with HBV-ACLF

Of the 539 patients with HBV-ACLF, 317 cases (58.81%) presented infection during the study, including 92 individuals who were infected at admission. The most common infection was SBP, followed by UTI, pulmonary infection and cholecystitis. Other rare infections included appendicitis (one case), perianal abscess (one case), cytomegalovirus (one case), herpes zoster (two cases), viral conjunctivitis (one case) and EB virus (one case). The distribution and incidence of infections were shown in Table 2.

Table 2 Prevalence and types of infection in patients with HBV-ACLF.

Types of infection	n	Prevalence of infection (%)	
SBP	170	31.54	
UTI	143	26.53	
Pulmonary infection	70	12.99	
Cholecystitis	35	6.49	
Other infections	7	1.30	
Note:

ACLF, acute-on-chronic liver failure; HBV, hepatitis B virus; SBP, spontaneous bacterial peritonitis; UTI, urinary tract infection.

The prevalence and characteristics of multiple infections in patients with HBV-ACLF

A total of 317 patients presented infection, but the overall prevalence of infections was extremely high (78.85%, 425/539). The results revealed that some patients suffered from multiple infections at the same time, and the most severe cases even had four types of infections, as shown in Fig. 1. The majority of patients presented bacterial infections, but a few had combined with viral infections. The specific distribution of multiple infections was shown in Table 3.

Figure 1 Prevalence of multi-organ infections in patients with HBV-ACLF.

Table 3 Specific distribution of multiple-organ infections in patients with HBV-ACLF.

	Infection sites	Cases of infection	
Two types of infections (n = 84)	SBP + UTI	36	
SBP + pulmonary infection	23	
UTI + pulmonary infection	8	
pulmonary infection + cholecystitis	5	
UTI + cholecystitis	5	
SBP + cholecystitis	4	
UTI + appendicitis	1	
SBP + EB virus	1	
Herpes zoster + viral conjunctivitis	1	
Three types of infections (n = 16)	SBP + UTI + pulmonary infection	14	
SBP + UTI + cholecystitis	1	
Pulmonary infection + UTI + cholecystitis	1	
Four types of infections (n = 2)	SBP + UTI + pulmonary infection + cholecystitis	1	
UTI + pulmonary infection + cytomegalovirus + perianal abscess	1	
Note:

ACLF, acute-on-chronic liver failure; HBV, hepatitis B virus; SBP, spontaneous bacterial peritonitis; UTI, urinary tract infection.

Characteristics of pathogens and antibiotic susceptibility test patterns in patients with infection

A total of 202 isolates were detected in patients with infection. Culture results revealed that the proportion of Gram-negative bacteria, Gram-positive bacteria and fungi were respectively 62.87% (127/202), 23.76% (48/202) and 13.37% (27/202). The most common organism that caused infection was E. coli 36.14% (73/202). K. pneumoniae 12.87% (26/202), E. faecium 12.37% (25/202), E. faecalis 8.91% (18/202), A. baumanii 4.95% (10/202) and P. aeruginosa 2.97% (6/202) were more moderately responsible for infections. Other 17 isolates of bacterial species included: C. freundii 0.99% (2/202), C. indologenes 0.5% (1/202), B. cepacia 0.5% (1/202), E. cloacae 0.5% (1/202), E. meningoseptica 0.5% (1/202), P. fluorescens 0.5% (1/202), P. mirabillis 0.5% (1/202), S. agalactiae 0.5% (1/202), S. maltophilia 1.49% (3/202), S. viridans 0.5% (1/202), S. aureus 0.99% (2/202), S. marcescens 0.5% (1/202) and S. Salivarius 0.5% (1/202). We also detected 27 fungi, including A. fumigatus 4.95% (10/202), C. albicans 3.96% (8/202), C. tropicalis 3.47% (7/202) and C. glabrata 0.99% (2/202). These results were shown in Fig. 2.

Figure 2 Characteristics of pathogens in patients with infection.

The antibiotic susceptibility test patterns showed that 89.76%, 81.1% and 77.95% of the Gram-negative isolates were sensitive to imipenem, tigecycline and piperacillin/tazobactam, respectively and that 65.35%, 59.06% and 51.97% of the isolates were resistant to piperacillin, cefotaxime, sulfamethoxazole and trimethoprim, respectively (Table 4). These results also showed that all the Gram-positive isolates were sensitive to linezolid, teicoplanin and vancomycin and that 50%, 45.83% and 41.67% of the isolates were resistant to penicillin G, ampicillin and levofloxacin, respectively (Table 5). Moreover, the antibiotic susceptibility test patterns also showed that 52.97% (107/202) of all pathogens were MDR bacteria and 4.95% (10/202) were XDR bacteria.

Table 4 Antibiotic susceptibility pattern of Gram-negative isolates in HBV-ACLF patients with infection.

S/I/R	Antibiotic discs tested (n = 127)	
CAZ n (%)	CTX n (%)	F n (%)	IPM n (%)	LEV n (%)	PRL n (%)	SCF n (%)	SXT n (%)	TGC n (%)	TZP n (%)	
S I R	78 (61.42) 9 (7.09) 40 (31.5)	46 (36.22) 6 (4.72) 75 (59.06)	88 (69.29) 16 (12.6) 23 (18.11)	114 (89.76) 5 (3.94) 8 (6.3)	61 (48.03) 3 (2.36) 63 (49.61)	32 (25.2) 12 (9.45) 83 (65.35)	82 (64.57) 27 (21.26) 18 (14.17)	57 (44.88) 4 (3.15) 66 (51.97)	103 (81.1) 17 (13.39) 7 (5.51)	99 (77.95) 18 (14.17) 10 (7.87)	
Note:

CAZ, Ceftazidime; CTX, Cefotaxime; F, Nitrofurantoin; IPM, Imipenem; LEV, Levofloxacin; PRL, Piperacillin; SCF, Cefoperazone/sulbactam; SXT, Sulfamethoxazole and trimethoprim; TGC, Tigecycline; TZP, Piperacillin/tazobactam; S, sensitive; I, intermediate; R, resistant.

Table 5 Antibiotic susceptibility pattern of Gram-positive isolates in HBV-ACLF patients with infection.

S/I/R	Antibiotic discs tested (n = 48)	
AMP n (%)	F n (%)	LEV n (%)	LZD n (%)	PG n (%)	TEC n (%)	TGC n (%)	VA n (%)	
S I R	25 (52.08) 1 (2.08) 22 (45.83)	20 (41.67) 10 (20.83) 18 (37.5)	27 (56.25) 1 (2.08) 20 (41.67)	48 (100)	22 (45.83) 2 (4.17) 24 (50)	48 (100)	41 (85.42) 4 (8.33) 3 (6.25)	48 (100)	
Note:

AMP, Ampicillin; F, Nitrofurantoin; LEV, Levofloxacin; LZD, Linezolid; PG, Penicillin G; TEC, Teicoplanin; TGC, Tigecycline; VA, Vancomycin; S, sensitive; I, intermediate; R, resistant.

The characteristics of anti-infection therapy in patients with HBV-ACLF

Among the 539 patients with HBV-ACLF, 516 cases received anti-infection therapy (95.73%). Of these patients, 27.83% received only a single anti-infective drug, while the others were involved with combined drugs (Fig. 3). Most patients received long-term anti-infection therapy during hospitalization, and even 36.36% of the patients were treated with anti-infective drugs for more than 1 month. These results were shown in Fig. 4. The average time of anti-infection therapy was 24.81 days. There was no significant difference in anti-infection treatment time between infection group and non-infection group (25.27 days vs 24.15 days, P = 0.946); however, the time was significantly prolonged when combined with multiple infections due to the duration and complex of presentation (48.56 days in triple infection and 50.5 days in quadruple infection).

Figure 3 The incidence of treatment with anti-infective drugs in patients with HBV-ACLF. (0 = no anti-infective drug; 1 = one anti-infective drug; 2 = two anti-infective drugs; 3 = three anti-infective drugs; 4 = four anti-infective drugs).

Figure 4 The time distribution of anti-infective therapy in patients with HBV-ACLF.

Anti-infective drugs can contain antibiotics, antiviral drug, or antifungal drugs, and we found that all infected patients with HBV-ACLF received antibiotics, including carbapenem (71.06%, 383/539), cephalosporin (56.03%, 302/539), penicillin + enzyme inhibitors (25.05%, 135/539), quinolones (22.82%, 123/539), glycopeptides (21.52%, 116/539) and tetracyclines (4.08%, 22/539). The incidence of antifungal and antiviral therapy separately was 7.42% (40/539) and 0.74% (4/539), respectively. The results were shown in Fig. 5.

Figure 5 The proportion of all anti-infective drugs in patients with HBV-ACLF.

Determinants of infections in patients with HBV-ACLF

We evaluated the association between clinical parameters and the risk of infections in patients with HBV-ACLF. The strongest associations were found between infection and age, male, taking alcohol, diuretics, HE, HS, cirrhosis, long-time bed, mechanical ventilation, PTA, ALT, ALB, TC, eGFR, Hb, PLT, MELD scores and ACLF grade. With infection as dependent variable and the above factors as covariates, taking alcohol, HE, HS, cirrhosis, albumin and eGFR were independent risk factors in the prevalence of infection in patients with HBV-ACLF. The determinants of risk factors for infection among individuals with HBV-ACLF were shown in Table 6.

Table 6 Logistic regression analysis (Forward: LR) of risk factors for infections in patients with HBV-ACLF.

Parameter	Univariable analysis	Multivariable model	
	OR (95% CI)	P value	OR (95% CI)	P value	
Age	1.035 [1.019–1.051]	0.000			
Male	0.558 [0.325–0.959]	0.035			
Drinker	1.479 [1.025–2.133]	0.036	1.728 [1.144–2.611]	0.009	
Diuretics	2.144 [1.46–3.147]	0.000			
HE	2.391 [1.53–3.737]	0.000	1.823 [1.087–3.058]	0.023	
HS	6.707 [3.272–13.75]	0.000	3.071 [1.381–6.829]	0.006	
Cirrhosis	2.227 [1.53–3.243]	0.000	1.734 [1.142–2.631]	0.01	
Long-time bed	1.739 [1.228–2.462]	0.002			
Mechanical ventilation	11.748 [1.546–89.244]	0.017			
PTA	0.972 [0.949–0.995]	0.018			
ALT	1 [0.999–1]	0.01			
ALB	0.868 [0.834–0.904]	0.000	0.878 [0.841–0.917]	P < 0.001	
TC	0.79 [0.649–0.962]	0.019			
eGFR	0.98 [0.973–0.987]	0.000	0.988 [0.981–0.996]	0.003	
Hb	0.984 [0.976–0.992]	0.000			
PLT	0.997 [(0.994–1]	0.024			
MELD scores	1.074 [1.034–1.116]	0.000			
ACLF grade					
Grade 1	Reference				
Grade 2	1.699 [1.111–2.598]	0.014			
Grade 3	2.543 [1.53–4.225]	0.000			
Note:

ACLF, acute-on-chronic liver failure; ALB, albumin; ALT, alanine aminotransferase; CI, confidence interval; eGFR, estimated glomerular filtration rate; Hb, hemoglobin; HBV, hepatitis B virus; HE, hepatic encephalopathy; HS, hepatorenal syndrome; MELD, model for end-stage liver disease; OR, odds ratio; PLT, platelet; PTA, prothrombin activity; TC, total cholesterol.

Prognosis of patients with HBV-ACLF who developed infection

In the infection group, the proportion of patients who were discharged after improvement was lower than that in non-infection group (χ2 = 34.984, P < 0.5). Meanwhile, the proportion of patients who were discharged automatically for financial, medical insurance or personal reasons was higher than that in non-infection group (χ2 = 25.171, P < 0.5). A total of 6.12% (33/539) patients died during hospitalization, and the mortality of patients with infection was higher (χ2 = 4.167, P < 0.5) The results were shown in Table 7.

Table 7 Outcomes of patients with HBV-ACLF in infection group and non-infection group.

Outcomes	Infection group (n = 317)	Non-infection group (n = 222)	χ2 value	P value	
Better discharge, n (%)	99 (31.23%)	126 (56.76%)	34.984	P < 0.001	
Automatic dischargea, n (%)	178 (56.15%)	76 (34.23%)	25.171	P < 0.001	
Liver transplantation, n (%)	6 (1.89%)	4 (1.8%)	0.006	1.000	
Coma, n (%)	9 (2.84%)	8 (3.6%)	0.25	0.617	
Death, n (%)	25 (7.89%)	8 (3.6%)	4.167	0.041	
Note:

a Including patients who were transferred to other hospitals, went home and gave up treatment.

Discussion

Patients with liver failure are often associated with massive necrosis of hepatocytes, impaired monocyte macrophage system and decreased immune function, which all can increase the risk and severity of infections (Cheruvattath & Balan, 2007). Many studies have shown that a variety of pathogens may cause infections in patients with liver failure, and some of patients can even have multiple infections at the same time (Leber, Spindelboeck & Stadlbauer, 2012; Cheruvattath & Balan, 2007; Fernández et al., 2018; Fernandez & Gustot, 2012; Shalimar et al., 2018). When the patients become complicated with infection, catabolism increases and toxic metabolic substances (including endotoxin and endogenous ammonia) cannot be fully removed, which causes much more damage to the liver and results in the further worsening of liver function (Leber, Spindelboeck & Stadlbauer, 2012). Therefore, in addition to encephaledema, infection is also often associated with significant mortality (Mucke et al., 2018; Wang et al., 2013).

Our research showed that 58.81% of patients with HBV-ACLF were complicated with infections. Among these patients, 17.07% (92/539) of patients were already infected at admission, while others developed a nosocomial infection. However, because our hospital was a higher-level referral unit, most of these patients who were infected at admission have been hospitalized in local hospitals. Of all types of infection, the most common was SBP, followed by UTI and pneumonia. Among the 317 infected patients, 32.18% (102/317) had at least two types of infection at the same time, and the most frequent sites of infection were the abdomen, the urinary tract, and the lungs. There may be several reasons for this. First, immune function may have decreased in patients with liver failure, accompanied by intestinal bacterial overgrowth and translocation. Altered permeability of the intestinal wall, disorder of intestinal function and the formation of the ascitic fluid could also cause intestinal infection. Once bacteria reached the ascitic fluid, it then became colonized, referred to as the seeding of ascitic fluid with blood-borne bacteria (Fernández et al., 2018). Second, during the decompensated period of liver failure, elevated portal pressure and portal systemic shunting made it easier for bacteria in the intestinal tract to enter pulmonary circulation. Bacteria would enter the blood circulation through the peritoneum and then colonized in the lungs, which led to pulmonary infection finally. Additionally, limited breathing movement due to the pressure of the built-up ascitic fluid can also cause pneumonia (Cheruvattath & Balan, 2007). Third, Hayes & Abraham (2016) have reported that a rapid and vigorous response mediated by the immune system was largely responsible for guarding against bacterial infections that bypass the natural defenses of the urinary tract. However, this capability can become greatly weakened in a state of immunological deficits such as that brought on by HBV-ACLF. Furthermore, the secondary release of norepinephrine at the intestinal mucosa can impair local immune system function and induce qualitative and quantitative changes of the intestinal microbiota towards a phenotype associated with bacterial translocation (Fernández et al., 2018). Indeed, evidence has shown that uropathogenic E. coli (UPEC) was responsible for more than 80% of UTIs and that this pathogen was believed to originate from the intestines as these bacteria could be regular components of the microbiome of the gastrointestinal tract (Hayes & Abraham, 2016; Ronald, 2003).

Research has also been reported that bacterial infection was a main precipitating factor in patients with ACLF and bacterial infections have also been found to be severe and associated with intense systemic inflammation, poor clinical courses and high mortality in patients with ACLF. Therefore, in our study, all infected patients with HBV-ACLF received anti-infective therapy according to antibiotic susceptibility patterns, and we found that E. coli was the most common causative organism of infection. Some studies also have shown that enterobacteria were the main pathogens of infection in patients with liver failure. However, it was worth noting that 95.73% of patients including the non-infective cases in this study received anti-infective therapy. Hence, in addition to the infection itself, we also took account the severity of disease and the general condition of patients. From this we found that the adequacy of empirical antibiotic strategies and antibiotic prophylaxis was a key factor in the management of infected patients with HBV-ACLF. Several authors have reported that inappropriate first-line therapies are associated with increased mortality and that the proper initial empiric antibiotic therapy can effectively reduce the mortality and hospitalization days of patients (Mucke et al., 2018; Fung, Lai & Yuen, 2014). In our study, Carbapenem and third generation cephalosporins were the drugs of first choice, and penicillin + enzyme inhibitors, quinolones and glycopeptides were also widely used. However, the increased usage of antibiotics was also associated with a clinically relevant and increasing drawback, the development of infections due to MDR organisms and fungi. In our study, antibiotic susceptibility test patterns showed that 52.97% of all bacteria found in pathogens were MDR bacteria and 4.95% were XDR bacteria. It might be because the patients usually comprised a high number of critically ill and some of them were given prophylactic antibiotics in local hospitals. This situation will further lead to poor treatment effect and prolonged hospitalization and anti-infection time. Therefore, the treating clinicians must have faced some controversy about whether to choose antibiotic prophylaxis. Several strategies have already been suggested to balance the risks and benefits of antibiotic prophylaxis. Some researchers have found that broad antibiotic schemes covering all potential pathogens should be applied at high doses within the first 48–72 h after the diagnosis of infection to try to improve clinical efficacy and minimize the selection of resistant strains (Fernández et al., 2018; Fernandez et al., 2016). However, the strategy in the department of infectious disease in this study was the use of antibiotics targeted at specific subpopulations at high risk for infection, such as those with prior episodes of spontaneous bacterial peritonitis, upper gastrointestinal bleeding, or low-protein ascites with associated poor liver function. Antifungal therapy was only used in patients who had several types of antibiotics and been using them for a long time. In addition, treating clinicians paid attention to the avoidance of antibiotic overuse and early de-escalation policies.

This study has also showed that the prognosis of infected patients was poor and the mortality was higher than patients without infection. Therefore, how to improve the long-term survival rate of infected patients is very important. In the study, multivariate logistic regression analysis showed that taking alcohol, HE, HS, cirrhosis, albumin and eGFR were risk factors for the development of infection in patients with HBV-ACLF. Alcohol and the products of its degradation by human organism can strongly modulate the human gut microbiota (Dubinkina et al., 2017). The sedative properties of alcohol which can reduce oropharyngeal tone, leading to an increased risk of aspiration of microbes. Furthermore, high levels of alcohol intake can modify alveolar macrophage function, hence diminishing pulmonary defence against infection. Also, high alcohol consumption is often associated with malnutrition, resulting in the impairment of immunity and increased infection risk (Simou, Britton & Leonardi-Bee, 2018). Cirrhosis was usually a background chronic liver disease upon which they developed acute insult resulting in ACLF. It was characterized by diffuse fibrosis, severe impairment of intrahepatic venous blood flow, portal hypertension and liver failure. Meanwhile, the onset of complications represented the beginning of decompensated cirrhosis, characterized by impaired liver function, as well as impairment of extrahepatic organs and systems, including HE (impaired cognitive, mental and motor functions) and HS (impaired renal sodium and water excretion, renal hemodynamics, renal perfusion and eGFR). It also meant the more severe tissue damage and higher levels of proinflammatory factors or chemokines, and “cytokine storm” further led to systemic inflammation, which aggravated infection (Ramachandran, Iredale & Fallowfield, 2015; Arroyo et al., 2016). We also found that albumin was a risk factor for the prognosis of liver failure and a predictor of morbidity and mortality. Damage of liver synthesis can cause a decrease in albumin, which can be used to assess the severity of dysfunction in the liver and immune system. Lower albumin has been found to be associated with malnutrition and weight loss, and can also exacerbate systemic inflammatory response and induce infection through clearing and regulating cytokine production (Gatta, Verardo & Bolognesi, 2012; Huang et al., 2017). In addition, eGFR has been found to be an indicator of renal function, and decreased eGFR might cause oliguria or anuria, azotemia and dilutional hyponatremia, which can all further induce hepatorenal syndrome, hepatic encephalopathy and endotoxemia, and aggravate infection finally (Moreau et al., 2013; Mackelaite, Alsauskas & Ranganna, 2009; Bucsics & Krones, 2017). Therefore, we should make best efforts to remove these risk factors that can aggravate infection in order to make patients to acquire better outcomes.

Our study still has several potential limitations, however. First, the study was only a single-center investigation. More data from large, multi-center studies are needed in the future to obtain more robust conclusions. Second, the follow-up data to evaluate the recurrence and complexity of infections in patients with HBV-ACLF were not available. Hence, further studies with multiple centers and larger samples should be conducted in the future.

In conclusion, patients with HBV-ACLF often complicated with infection that could aggravate liver failure and seriously affect their survival and prognoses. Anti-infection therapy is crucial in the treatment of liver failure and individualized management, including the adequacy of empirical antibiotic strategies and antibiotic prophylaxis was needed due to the complexity and difficulty of multiple possible infections, however, antibiotic overuse and prolonged hospitalization should be avoided. We found taking alcohol, HE, HS, cirrhosis, albumin and eGFR to be independent risk factors in the prevalence of infection in patients with HBV-ACLF. Therefore, it was necessary to reduce alcohol consumption, prevent the occurrence of complications, correct hypoalbuminemia and protect renal function using multimodal therapy for liver failure. Moreover, early detection of pathogenic bacteria, prompt treatment, positive prevention of infections and early de-escalation of antibiotics may help to improve the quality of life and survival rate in patients with HBV-ACLF. Our findings add to the epidemiology profile of infections across the spectrum of HBV-ACLF and may be beneficial in future management in patients of HBV-ACLF.

Supplemental Information

Supplemental Information 1 Data of the patients with HBV-ACLF.

Click here for additional data file.

Supplemental Information 2 Checklist.

Click here for additional data file.

Supplemental Information 3 Instructions for categorical data.

Click here for additional data file.

The authors are indebted to Zheng-ce Wan and Li Liu for assistance with the methodology. The authors thank AiMi Academic Services for the English language editing and review services.

Additional Information and Declarations

Competing Interests

Author Contributions

Human Ethics

Data Availability

The authors declare that they have no competing interests.

Qian Zhang conceived and designed the experiments, performed the experiments, analyzed the data, prepared figures and/or tables, authored or reviewed drafts of the article, and approved the final draft.

Baoxian Shi performed the experiments, analyzed the data, prepared figures and/or tables, and approved the final draft.

Liang Wu conceived and designed the experiments, authored or reviewed drafts of the article, and approved the final draft.

The following information was supplied relating to ethical approvals (i.e., approving body and any reference numbers):

The Ethics Research Committee at Tongji Hospital approved the study (TJ-IRB20190601).

The following information was supplied regarding data availability:

The data are available in the Supplemental File.

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
