# Peer review of "Characteristics and risk factors of infections in patients with HBV-related acute-on-chronic liver failure: a retrospective study"

_PeerJ, doi:10.7717/peerj.13519_

## Round 0.1 · original submission · Major Revisions

Your manuscript has been reviewed and assessed by two reviewers, and both of them agree with the fact that there are still a few points that need to be addressed. The comments of the reviewers are included at the bottom of this letter. Reviewers indicated that methods, results, and discussion sections should be improved. Reviewers also recommended extensive English editing. We would be glad to consider a substantial revision of your work, where the reviewer’s comments will be carefully addressed one by one. In addition to these, please see my comments below:

-In abstract: Please change “Univariate and Logistic regression analyses” to “Univariate Logistic regression analyses”. It may be better to add this sentence to the abstract: “Differences among groups were compared with Student’s t-test and Mann Whitney U test. Univariate logistic regression analysis was used for modeling the relationship between infection and clinical characteristics.”

- The methods section needs more elaboration, in particular, data analysis. Despite the need for multivariate analysis, the authors performed logistic regression analysis only in a univariate manner. Univariate analysis can be very risky in situations where various confounding effects affect the outcome. Please provide multivariate analysis results in Table 4 and update the methods section. Give details of the multivariate analysis results (stepwise/backward etc.) How did you check the normality assumption of the data? Please give the name of the statistical test for normality.

- Hypotheses were not clear, why this study was very important to address? What was the gap in knowledge? Give these details in the last paragraph of the introduction and provide the purpose of the study in this paragraph.

- Line 118: Add “IBM” before “SPSS 20.0 software”.

- What was the power of the study? Which statistical package or tool was used to calculate sample size? Please provide the name of the package or the tool.

-In Table 3: I’m a little confused. If the data are not normally distributed, the authors should give the median and IQR values for the Mann-Whitney U test (not mean and standard deviation), and also provide the name of the statistical test as annotation of the table (a Student t-test, b Mann-Whitney U test, etc.). If this table is made for univariate logistic regression, update the title and provide OR and p-values the same as Table 4. Also, share your group differences in another table. Because you mentioned that “Differences among groups were compared with Student’s t-test and Mann Whitney U test.”.

Reviewer 1 ·

Basic reporting

See below

Experimental design

See below

Validity of the findings

This is a a retrospective analysis of 539 patients with HBV-ACLF. The manuscript is well written but lack of innovation. The data presented do not bring new messages to the current knowledge.

Major concerns:

1- The authors used the APASL classification of ACLF that lacks data on ACLF grading.
The EASL-CLIF classification allow grading of ACLF by severity. We might except that infection rates was higher in patients admitted with ACLF 3. Please heck if you can classify the patients by ACLF grading at time of admission

2-Other scores like the Child-Pugh score, the MELD score could allow better classification of infection occurrence according to liver severity.

3-Unless I missed, it's not clear whether the patients were admitted in the ward or in ICU

4- According to above, how many patients under mechanical ventilation or underwent dialysis ?

5- Type of Bacterial infection and strains identification, resistance phenotype are not reported ?

6- Carbapenem was the most common antibiotic used. How many patients had BMR or MDRO ?

7- What was the proportion of patients who had community acquired infection and those who developed a nosocomial infection ?

8- It's not reported what incidence and type of fungal and viral infection.

9- Outcome of patients who developed infection ?

Additional comments

None

Reviewer 2 ·

Basic reporting

The authors describe the prevalence and predictors of infections in patients with HBV ACLF. However, there are certain flaws in the manuscript in its present form as highlighted below
1. English language is very poor and requires proof reading by an English expert
2. Background shows context but needs some literature review on the topic
3. Raw data has not been supplied
4. Figures are well made but are a replication of what is mentioned in the text

Experimental design

1. A major flaw of this study is the inclusion of all patients with infection. Infection may also have been the precipitant of ACLF
2. Ideally the authors should have only included patients who were non-infected at admission and then developed infection after 48 hours of admission to evaluate the risk and predictors of infection
3. The authors mention cirrhosis as a severe complication of ACLF, but cirrhosis is usually a background chronic liver disease in these patients upon which they develop acute insult resulting in ACLF. Authors should clarify this
4. Acute cholecystitis may be an infectious complication if acalculous. However, GB wall thickening may also be a manifestation of acute hepatitis
5. Acute pancreatitis is not an infection, authors should clarify this
6. The authors should provide reference for the HBV DNA cut offs.
7. Were these patients HBe Ag negative and positive or HBs Ag negative and positive ?
8. The authors should clarify, why some patients received antibiotic for more than 1 month
9. The lines 159 – 161 in the results section should have come in the beginning of results section
10. What was the acute precipitant of ACLF in these patients ?
11. What was the duration of presentation in these patients
12. How many patients had index presentation as ACLF ?
13. As per the canionic definition, what were the ACLF grades in these patients

Validity of the findings

1. Discussion needs to refer more studies on infection and ACLF (Dig Liver Dis. 2018 Nov;50(11):1225-1231)
2. Relevant findings but not adequately represented

Additional comments

None

---

## Round 0.2 · Minor Revisions

Your manuscript has been reviewed and assessed by two reviewers, and one of them recommended adding a new reference. I also have concerns as below:

-Line 386: Please correct “Shapiro Wilk test”

-Please provide power analysis details of the study in the methods section: What was the power of the study (give the results of priori and posteriori power analysis)? Which statistical package or tool was used to calculate the sample size? What was the effect size and which references were used in the calculation?

-Correct p-values of “0.000” to “p<0.001” in tables.

Reviewer 1 ·

Basic reporting

Clear

Experimental design

Not applicable

Validity of the findings

Yes

Additional comments

None

Reviewer 2 ·

Basic reporting

Fine

Experimental design

Fine

Validity of the findings

Fine

Additional comments

The paper has been revised adequately.
Please cite the following reference.
Shalimar, Rout G, Jadaun SS, Ranjan G, Kedia S, Gunjan D, Nayak B, Acharya SK, Kumar A, Kapil A. Prevalence, predictors and impact of bacterial infection in acute on chronic liver failure patients. Dig Liver Dis. 2018 Nov;50(11):1225-1231. doi: 10.1016/j.dld.2018.05.013.

---

## Round 0.3 · accepted · Accept

The authors addressed the reviewers' concerns and substantially improved the content of the manuscript. So, based on my own assessment as an academic editor, no further revisions are required and the manuscript can be accepted in its current form.